# Quantum quench dynamics in the transverse-field Ising model: A numerical expansion in linked rectangular clusters

**Jonas Richter⋆, Tjark Heitmann and Robin Steinigeweg**

Department of Physics, University of Osnabrück, D-49069 Osnabrück, Germany

⋆ jonasrichter@uos.de

## Abstract

We study quantum quenches in the transverse-field Ising model defined on different lattice geometries such as chains, two- and three-leg ladders, and two-dimensional square lattices. Starting from fully polarized initial states, we consider the dynamics of the transverse and the longitudinal magnetization for quenches to weak, strong, and critical values of the transverse field. To this end, we rely on an efficient combination of numerical linked cluster expansions (NLCEs) and a forward propagation of pure states in real time. As a main result, we demonstrate that NLCEs comprising solely rectangular clusters provide a promising approach to study the real-time dynamics of two-dimensional quantum many-body systems directly in the thermodynamic limit. By comparing to existing data from the literature, we unveil that NLCEs yield converged results on time scales which are competitive to other state-of-the-art numerical methods.



# 1 Introduction

Understanding the dynamics of isolated quantum many-body systems out of equilibrium is an active area of research of modern theoretical and experimental physics [1–3]. A popular nonequilibrium protocol in this context is a so-called quantum quench [4]. In such quench protocols, the system's Hamiltonian $\mathcal{H}$ depends on some parameter $\lambda$, and the system is prepared in an eigenstate $|\psi(0)\rangle$ of $\mathcal{H}$, e.g., the groundstate, for an initial value $\lambda_i$. Next, the value of $\lambda$ is suddenly changed, $\lambda_i \rightarrow \lambda_f$, such that $|\psi(0)\rangle$ is no eigenstate of $\mathcal{H}(\lambda_f)$, and the system exhibits nontrivial dynamics. For an isolated quantum system undergoing unitary time evolution, it is then intriguing to study if and in which way the system relaxes back to equilibrium. Central questions are, for instance, how the (short- or long-time) dynamics can be described in terms of "universal" principles [1, 5–11], what are the relevant time scales of relaxation [12–14], and whether or not the long-time values of physical observables agree with the prediction of, e.g., a microcanonical or canonical ensemble (i.e. thermalization) [15–17].

One possible mechanism to explain the emergence of thermalization in isolated quantum systems is given by the eigenstate thermalization hypothesis (ETH) [18–20]. While the validity of the ETH has been numerically tested for a variety of models and observables (see, e.g., [21–27]), there are also classes of systems which violate the ETH and fail to thermalize. One such class is given by integrable models, where the extensive number of conservation laws prevents the applicability of standard statistical ensembles [28]. Instead, it has been proposed that integrable models equilibrate towards a generalized Gibbs ensemble (GGE), which maximizes the entropy with respect to the conserved charges [29–31]. In addition, it is now widely believed that some strongly disordered systems can undergo a transition to a many-body localized (MBL) phase, where the ETH is violated as well [32, 33]. Moreover, there has been plenty of interest recently in models which are, in a sense, intermediate cases between "fully ETH" and "fully MBL". This includes, e.g., models featuring "quantum scars" where rare ETH-violating states are embedded in an otherwise thermal spectrum [34–38], as well as models which exhibit a strong fragmentation of the Hilbert space due to additional contraints [39, 40].

From a numerical point of view, studying the nonequilibrium dynamics of isolated quantum many-body systems is a challenging task. This is not least caused by the fact that for an interacting quantum system, the Hilbert space grows exponentially in the number of constituents. Nevertheless, thanks to the continuous increase of computational resources and the development of sophisticated numerical methods including, e.g., dynamical mean field theory [41], Krylov subspace techniques [42, 43], dynamical quantum typicality [44], or classical representations in phase space [45], significant progress has been made. Especially for one-dimensional systems, the time-dependent density-matrix renormalization group, as well as related methods based on matrix-product states (MPS), provide a powerful tool to study dynamical properties for system sizes practically in the thermodynamic limit [46, 47]. However, since these methods rely on an efficient compression of moderately entangled wavefunctions, the reachable time scales in simulations are eventually limited due to the inevitable buildup

of entanglement during the unitary time evolution.

The growth of entanglement becomes even more severe in spatial dimensions larger than one. Despite recent advances involving MPS-based or tensor-network algorithms [48–53], as well as the advent of innovative machine-learning approaches [54–56], the time scales numerically attainable for two-dimensional quantum many-body systems are still comparatively short. While the dynamics of such two-dimensional systems can nowadays be accessed in experiments with quantum simulators [57–59], the development of efficient numerical techniques is paramount. On the one hand, unbiased numerical simulations are important to confirm the accuracy of the experimental results. On the other hand, numerical simulations can also serve as an orientation for experiments to explore certain models or parameter regimes in more detail.

In this paper, we scrutinize the nonequilibrium dynamics for quantum quenches in the Ising model with transverse magnetic field. While this model is exactly solvable in the case of a chain and has been studied in numerous instances, our main focus is on nonintegrable geometries such as two- and three-leg ladders and, in particular, two-dimensional square lattices. To this end, we rely on an efficient combination of numerical linked cluster expansions (NLCEs) and the iterative forward propagation of pure states in real time via Chebyshev polynomials. Depending on the model geometry, the initial state, and the strength of the quench, the nonequilibrium dynamics is found to display a variety of different behaviors ranging from rapid equilibration, over slower monotonous relaxation, to persistent (weakly damped) oscillations. Most importantly, from a methodological point of view, we demonstrate that NLCEs comprising solely rectangular clusters provide a promising approach to study the real-time dynamics of two-dimensional quantum many-body systems directly in the thermodynamic limit. By comparing to existing data from the literature, we unveil that NLCEs yield converged results on time scales which are competitive to other state-of-the-art numerical methods.

This paper is structured as follows. In Sec. 2, we introduce the models, observables, and quench protocols which are studied. In Sec. 3, we then discuss the employed numerical methods, while our results are presented in Sec. 4. We summarize and conclude in Sec. 5.

## 2  Models, observables, and quench protocols

We study the Ising model with ferromagnetic nearest-neighbor interactions and transverse magnetic field, described by the Hamiltonian

$$\mathcal{H} = -J \left( \sum_{\langle \ell,m \rangle} \sigma_\ell^z \sigma_m^z + g \sum_{\ell=1}^{L} \sigma_\ell^x \right), \tag{1}$$

where the first sum on the right hand side runs over all pairs of nearest neighbors $\ell$ and $m$, $L$ is the total number of sites, $J > 0$ sets the energy scale, $g > 0$ denotes the strength of the transverse field, and $\sigma_\ell^{x,z}$ are Pauli matrices at site $\ell$. Note that the Hamiltonian (1) is symmetric under the global spin-flip operation $\sigma_\ell^z \to -\sigma_\ell^z$.

In this paper, the transverse-field Ising model (1) is considered for different lattice geometries such as chains ($L = L_x$), two- and three-leg ladders ($L = L_x \times 2, L = L_x \times 3$), and two-dimensional square lattices ($L = L_x \times L_y$). While we generally intend to obtain results in the thermodynamic limit $L \to \infty$ (see Sec. 3.1 for our numerical approach), we consider finite system sizes as well. In the case $L < \infty$, one has to distinguish between open boundary conditions (OBC) and periodic boundary conditions (PBC), where for chains and ladders the latter only applies in the $x$ direction.

On the one hand, in the case of a chain, $\mathcal{H}$ is a paradigmatic example of an integrable model and can be solved exactly by subsequent Jordan-Wigner, Fourier, and Bogolioubov trans-

forms [60], see also Appendix A. For $g < 1$, $\mathcal{H}$ is in a ferromagnetic phase with a two-fold degenerate groundstate. At the critical point $g = 1$, $\mathcal{H}$ undergoes a quantum phase transition towards a paramagnetic phase with unique groundstate for $g > 1$. On the other hand, for a two-dimensional square lattice, $\mathcal{H}$ is nonintegrable [24, 25, 61], and the quantum phase transition between an ordered phase and an unordered phase occurs at the larger transverse field $g = g_c \approx 3.044$ [62]. For intermediate cases, such as multi-leg ladders on a cylinder geometry, the value of $g_c$ can vary since these cases are quasi-one-dimensional [50].

In this paper, we consider quench protocols starting from fully polarized initial states $|\psi(0)\rangle$. Namely, we either study quenches starting from $|\psi(0)\rangle = |\uparrow\rangle$,

$$|\uparrow\rangle = |\uparrow\uparrow\cdots\uparrow\rangle \, , \tag{2}$$

where all spins are initially aligned along the $z$ axis, or quenches starting from the state $|\psi(0)\rangle = |\rightarrow\rangle$,

$$|\rightarrow\rangle = |\rightarrow\rightarrow\cdots\rightarrow\rangle \, , \tag{3}$$

where all spins point in the $x$ direction. Note that written in the common eigenbasis of the local $\sigma_\ell^z$, $|\rightarrow\rangle$ is a uniform superposition of all $2^L$ basis states. Moreover, while the state $|\uparrow\rangle$ is an eigenstate of $\mathcal{H}$ for vanishing field $g = 0$, the state $|\rightarrow\rangle$ is the groundstate of $\mathcal{H}$ for $g \to \infty$. Given the states $|\uparrow\rangle$ and $|\rightarrow\rangle$, we study the nonequilibrium dynamics resulting from quantum quenches to weak ($g < g_c$), strong ($g > g_c$), or critical values ($g = g_c$) of the transverse field, i.e., depending on the initial state these are quenches either within the same equilibrium phase, or to or across the critical point.

Due to the quench, the fully polarized states $|\uparrow\rangle$ and $|\rightarrow\rangle$ are no eigenstates of $\mathcal{H}$ anymore and evolve unitarily in time ($\hbar = 1$),

$$|\psi(t)\rangle = e^{-i\mathcal{H}t}|\psi(0)\rangle \, . \tag{4}$$

Consequently, the expectation values of observables acquire a dependence on time as well. In particular, we here consider the dynamics of the transverse and the longitudinal magnetization,

$$\langle X(t)\rangle = \frac{1}{L}\sum_{\ell=1}^{L}\langle\psi(t)|\sigma_\ell^x|\psi(t)\rangle \, , \quad \langle Z(t)\rangle = \frac{1}{L}\sum_{\ell=1}^{L}\langle\psi(t)|\sigma_\ell^z|\psi(t)\rangle \, . \tag{5}$$

## 3 Numerical approach

We now discuss the numerical methods which are employed in this paper. Throughout this section, we exemplarily focus on the transverse magnetization $\langle X(t)\rangle$. The calculations for $\langle Z(t)\rangle$ are carried out analogously.

### 3.1 Numerical linked cluster expansion

Numerical linked cluster expansions provide a means to access the properties of quantum many-body systems directly in the thermodynamic limit. Originally introduced to study thermodynamic quantities [63, 64] (see also [65–67]), NLCEs have more recently been employed to obtain entanglement entropies [68], to calculate steady-state properties in driven-dissipative systems [69], to study quantum quenches with mixed or pure initial states [58, 70–74], as well as to simulate time-dependent equilibrium correlation functions [75, 76].

The main idea of NLCEs is that the per-site value of an extensive quantity in the thermodynamic limit can be obtained as a sum over contributions from all linked clusters which can be embedded on the lattice [77],

$$\lim_{L\to\infty} \langle X(t)\rangle = \sum_c \mathcal{L}_c W_c(t)\,, \tag{6}$$

where the sum runs over all connected clusters $c$ with multiplicities $\mathcal{L}_c$ and weights $W_c(t)$. Specifically, $\mathcal{L}_c$ is the number of ways (normalized by the size of the lattice) a cluster $c$ can be embedded on the lattice [see also the discussion around Eq. (9) below]. Moreover, the notion of a *connected* cluster refers to a finite number of lattice sites, where every site of the cluster has to be directly connected to at least one other cluster site by terms of the underlying Hamiltonian. Given a two-dimensional square lattice and the nearest-neighbor Hamiltonian in Eq. (1), for instance, the lattice sites $(i,j)$ and $(i,j+1)$ form a connected cluster of size two. In contrast, the sites $(i,j)$ and $(i+1,j+1)$ do not form a connected cluster as $\mathcal{H}$ does not contain terms along the diagonal. However, in combination, the sites $(i,j)$, $(i,j+1)$, and $(i+1,j+1)$ would be a connected cluster of size three.

Given a cluster $c$, its weight $W_c(t)$ is obtained by an inclusion-exclusion principle. That is, the quantity of interest (here the dynamics of the magnetization $X$) is evaluated on the cluster $c$ (with OBC) and, subsequently, the weights $W_s(t)$ of all subclusters $s$ of $c$ have to be subtracted [77],

$$W_c(t) = \langle X(t)\rangle_{(c)} - \sum_{s\subset c} W_s(t)\,. \tag{7}$$

While NLCEs yield results in the thermodynamic limit (such that a finite-size scaling becomes unnecessary), it is instead crucial to check the convergence of the series. To this end, the sum in Eq. (6) is usually organized in terms of expansion orders [77]. For instance, one could group together all clusters which comprise a certain number of lattice sites. Then, an expansion up to order $C$ refers to the fact that all clusters with up to $C$ lattice sites are considered in Eq. (6). Moreover, the NLCE is said to be converged if the outcome of Eq. (6) does not depend on the value of $C$.

At this point, it is important to note that in actual simulations, the maximum order $C$ that can be reached is limited by two factors: (i) the exponential growth of the Hilbert-space dimension with increasing cluster size, and (ii) the necessity to identify the (possibly very large number of) distinct clusters and to calculate their weights. Since a larger expansion order typically leads to a convergence of Eq. (6) up to longer times [75] (or down to lower temperatures for thermodynamic quantities [66]), it is desirable to include clusters as large as possible. In this paper, we therefore aim to mitigate the limitations (i) and (ii) by two complementary approaches. First, instead of using full exact diagonalization to evaluate $\langle X(t)\rangle_{(c)}$, we here employ an efficient forward propagation of pure states (see Sec. 3.2), which is feasible for significantly larger Hilbert-space dimensions. Secondly, in order to reduce the enormous combinatorial costs to generate (and evaluate) all clusters with a given number of sites, we rely on the fact that the sum in Eq. (6) can also converge for different types of expansions, as long as clusters and subclusters can be defined in a self-consistent manner [77]. In this paper, we specifically restrict ourselves to only those clusters which have a rectangular shape. This restriction is particularly appealing as the number of distinct clusters is significantly reduced and the calculation of the weights $W_c(t)$ becomes rather simple since all subclusters are rectangles as well, see Fig. 1. Furthermore, the rectangle expansion has been succesfully used before to obtain entanglement entropies [68], and it also appears to be a promising candidate to study dynamical properties as it involves clusters with many different length scales. In this context, let us note that other restricted expansions for the two-dimensional square lattice, e.g., clusters consisting of corner-sharing $2\times 2$ squares, have proven to be a good choice to ex-

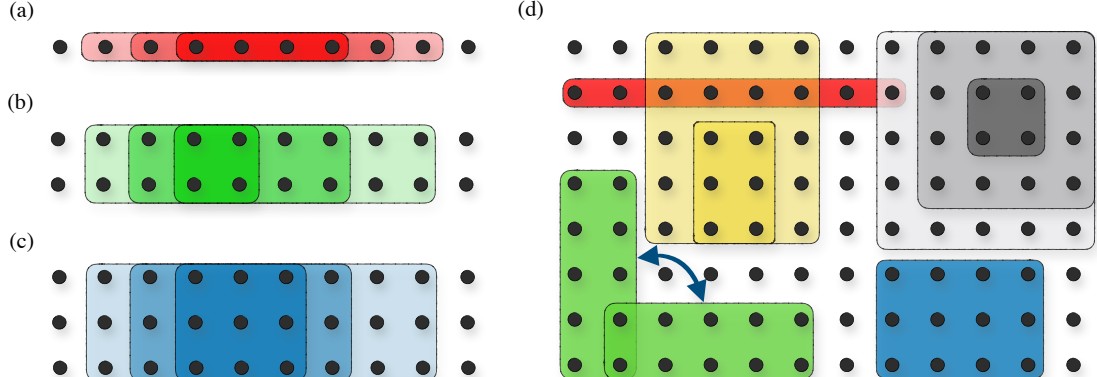

Figure 1: Examples of clusters which are used in the NLCE. (a) For a chain geometry, all clusters and subclusters are chains. (b) and (c) In case of a ladder geometry, we only consider clusters and subclusters which are ladders as well. (d) For the two-dimensional square lattice, we restrict ourselves to clusters with a rectangular shape. Given the Hamiltonian $\mathcal{H}$ in Eq. (1), we note that a cluster $c = (x, y)$ with $x > y$ is equivalent to its 90°-rotated counterpart $c' = (y, x)$. To speed up the simulations, we therefore only need to consider clusters $c$ with $x \geq y$, where square-shaped clusters with $x = y$ enter Eq. (6) once, while rectangular clusters with $x > y$ enter the expansion twice.

tract thermodynamic quantities [66]. In this paper, however, we focus on rectangular clusters as a first case study.

Given a rectangular cluster $c = (x, y)$ of width $x$ and height $y$, the inclusion-exclusion principle from Eq. (7) to obtain the weight $W_{(x,y)}(t)$ takes on the form [78]

$$W_{(x,y)}(t) = \langle X(t) \rangle_{(x,y)} - \sum_{x'=1}^{x} \sum_{\substack{y'=1 \\ x'y'<xy}}^{y} (x - x' + 1)(y - y' + 1) W_{(x',y')} \,, \tag{8}$$

where the sum runs over all rectangular subclusters. Next, in order to carry out the expansion (6), the multiplicity $\mathcal{L}_c$ is required. Given a two-dimensional square lattice of size $L_x \times L_y$ with OBC, the number of ways per lattice site a rectangular cluster $c = (x, y)$ (with finite $x$ and $y$) can be embedded on the lattice follows as,

$$\mathcal{L}_c = \frac{(L_x - x + 1)(L_y - y + 1)}{L_x L_y} \,, \tag{9}$$

as there are $(L_x - x + 1)$ possible translations in the $x$ direction and $(L_y - y + 1)$ in the $y$ direction. Thus, if one is interested in the properties of the lattice in the thermodynamic limit, $L_x, L_y \to \infty$, one finds,

$$\mathcal{L}_c = 1 \,. \tag{10}$$

Furthermore, in order to speed up the simulations, it is useful to take into account that the Hamiltonian $\mathcal{H}$ in Eq. (1) is invariant under rotations in the sense that a rectangular cluster $c = (x, y)$ with $x \geq y$ yields the exact same weight $W_c(t)$ as the cluster $c' = (y, x)$, i.e., $c$ rotated by 90 degrees. Thus, in practice, we only need to consider clusters with $x \geq y$, where square-shaped cluster with $x = y$ enter Eq. (6) once, while rectangular clusters with $x > y$ enter the expansion twice.

Let us add some comments on the NLCE for chains and ladders. First, we note that in the case of chains, all clusters are just chains as well, see Fig. 1 (a). In this case, the expansion in Eq. (6) reduces to a single difference between $\langle X(t)\rangle_{(c)}$ evaluated on the largest and the second-largest cluster [75]. Secondly, while for two-leg (or three-leg) ladders, rectangular clusters can in principle have a height $y = 1, 2$ (or $y = 1, 2, 3$) with different lengths $x$, we here restrict ourselves even further to those clusters which are two-leg or three-leg ladders as well, see Figs. 1 (b) and (c). In this case, the expansion (6) again reduces to a single difference between $\langle X(t)\rangle_{(c)}$ evaluated on the largest and the second-largest cluster. Despite this simplicity, however, we find that this type of expansion for ladders in practice yields convincing convergence times.

## 3.2 Pure-state propagation

Evaluating the unitary time evolution of the initial states $|\psi(0)\rangle$ according to Eq. (4) in principle requires the full exact diagonalization (ED) of the Hamiltonian $\mathcal{H}$. In order to access system (and cluster) sizes beyond the range of full ED, we here subdivide the evolution up to time $t$ into a product of discrete time steps,

$$|\psi(t)\rangle = e^{-i\mathcal{H}t}|\psi(0)\rangle = \left(e^{-i\mathcal{H}\delta t}\right)^Q |\psi(0)\rangle \,, \tag{11}$$

where $\delta t = t/Q$. If the time step $\delta t$ is chosen sufficiently small, then there exist various approaches to accurately approximate the action of the exponential $\exp(-i\mathcal{H}\delta t)$ such as, e.g., Trotter decompositions [79], Krylov subspace techniques [42], or Runge-Kutta schemes [80, 81]. In this paper, we rely on an expansion of the time-evolution operator in terms of Chebyshev polynomials, for a comprehensive overview see [82–85]. Let us emphasize that the evaluation of Eq. (11) to a high precision is crucial for the convergence of the NLCE. Even relatively small numerical errors for the contribution of each individual cluster could eventually spoil the convergence of the series when combined according to Eq. (6). In this context, the Chebyshev-polynomial expansion is known to yield very accurate results for a given step size $\delta t$. In contrast, we have checked that reaching the same level of accuracy by means of a fourth-order Runge-Kutta scheme requires a significantly smaller $\delta t$ which, in turn, increases the overall runtime of the simulation.

Since the Chebyshev polynomials are defined on the interval $[-1, 1]$, the spectrum of the original Hamiltonian $\mathcal{H}$ has to be rescaled [85],

$$\widetilde{\mathcal{H}} = \frac{\mathcal{H} - b}{a} \,, \tag{12}$$

where $a$ and $b$ are suitably chosen parameters. In practice, we use the fact that the (absolute of the) extremal eigenvalue of $\mathcal{H}$ can be bounded from above according to [83]

$$\max(|E_{\min}|, |E_{\max}|) \leq J\left(N_{\langle \ell, m\rangle} + gL\right) = \mathcal{E} \,, \tag{13}$$

where $E_{\max}$ ($E_{\min}$) is the largest (smallest) eigenvalue of $\mathcal{H}$, and $N_{\langle \ell, m\rangle}$ denotes the number of nearest-neighbor pairs $\langle \ell, m\rangle$, i.e., the number of bonds of the lattice. By choosing $a \geq \mathcal{E}$, it is guaranteed that the spectrum of $\widetilde{\mathcal{H}}$ lies within $[-1, 1]$. As a consequence, we can set $b = 0$. Note that while this choice of $a$ and $b$ is not necessarily optimal, it proves to be sufficient [83] (see also Appendix B).

Within the Chebyshev-polynomial formalism, the time evolution of a state $|\psi(t)\rangle$ can then be approximated as an expansion up to order $M$ [85],

$$|\psi(t + \delta t)\rangle \approx c_0 |v_0\rangle + \sum_{k=1}^{M} 2c_k |v_k\rangle \,, \tag{14}$$

where the expansion coefficients $c_0, c_1, \ldots, c_M$, are given by

$$c_k = (-i)^k \mathcal{J}_k(a\delta t) , \tag{15}$$

with $\mathcal{J}_k(a\delta t)$ being the $k$-th order Bessel function of the first kind evaluated at $a\delta t$. [Note that the notation in Eqs. (14) and (15) assumes $b = 0$.] Moreover, the vectors $|v_k\rangle$ are recursively generated according to

$$|v_{k+1}\rangle = 2\widetilde{\mathcal{H}} |v_k\rangle - |v_{k-1}\rangle , \quad k \geq 1 , \tag{16}$$

with $|v_1\rangle = \widetilde{\mathcal{H}} |v_0\rangle$ and $|v_0\rangle = |\psi(t)\rangle$. Given a time step $\delta t$ (and the parameter $a$), the expansion order $M$ has to be chosen large enough to ensure negligible numerical errors. In this paper, we typically have $\delta t J = 0.02$ and $M = 15$, which turns out to yield very accurate results (see Appendix B).

As becomes apparent from Eqs. (14) and (16), the time evolution of the pure state $|\psi(t)\rangle$ requires the evaluation of matrix-vector products. Since $\widetilde{\mathcal{H}}$ is a sparse matrix, these matrix-vector multiplications can be implemented comparatively time and memory efficient. In particular, we here calculate the matrix elements of $\widetilde{\mathcal{H}}$ *on the fly* and use parallelization to reduce the runtime. Thus, the memory requirements are essentially given by the size of the state $|\psi(t)\rangle$ and the auxiliary states $|v_{k-1}\rangle$, $|v_k\rangle$, and $|v_{k+1}\rangle$. As a consequence, it is possible to treat system (or cluster) sizes significantly larger compared to full ED (here up to 28 lattice sites with a Hilbert-space dimension of $d \approx 10^8$). Since the transverse-field Ising model (1) does not conserve the total magnetizations $X$ or $Z$, the corresponding quantum numbers cannot be used to block-diagonalize $\mathcal{H}$. Moreover, the clusters entering the NLCE are defined with open boundary conditions such that translational invariance cannot be exploited. Let us note that the clusters do have a reflection (parity) symmetry, which in principle can be used to reduce the memory requirements (though the reduction is less strong compared to the other symmetries mentioned before). In this paper, however, we do not exploit the reflection symmetry and always work in the full Hilbert space with dimension $d = 2^L$.

# 4 Results

We now present our numerical results for the quench dynamics of $\langle X(t)\rangle$ and $\langle Z(t)\rangle$ in chains, ladders, and two-dimensional lattices. Our main focus is to analyze the convergence properties of the NLCE by comparing to direct simulations of finite systems with periodic boundary conditions (for open boundary conditions, see Appendix C) and to existing data from the literature.

## 4.1 Chains

The transverse-field Ising chain is a paradigmatic example of an exactly solvable model and analytical solutions have been known for a long time [60, 86–88] (see also Appendix A). Since quantum quenches in the Ising chain have been studied extensively before (see, e.g., Refs. [89–96]), the present section should be mainly understood as a consistency check for our numerical methods and a preparation for the study of ladders and two-dimensional lattices in Secs. 4.2 and 4.3. (It might be fair to say, however, that explicit visualizations of the analytical solutions, e.g., for the full time-dependent relaxation process of $\langle X(t)\rangle$ for specific initial states and transverse fields $g$, are less often available in the literature.)

In Figs. 2 (a)-(c), the dynamics of the transverse magnetization $\langle X(t)\rangle$ is shown for quenches starting from the initial state $|\psi(0)\rangle = |{\uparrow}\rangle$ and different values of the transverse field $g = 0.5, 1, 2$. (Recall that the quantum critical point is $g = 1$ for the chain geometry.) Numerical data obtained by NLCE for expansion orders $C = 24, 25$ are compared to (i) a simulation for a finite

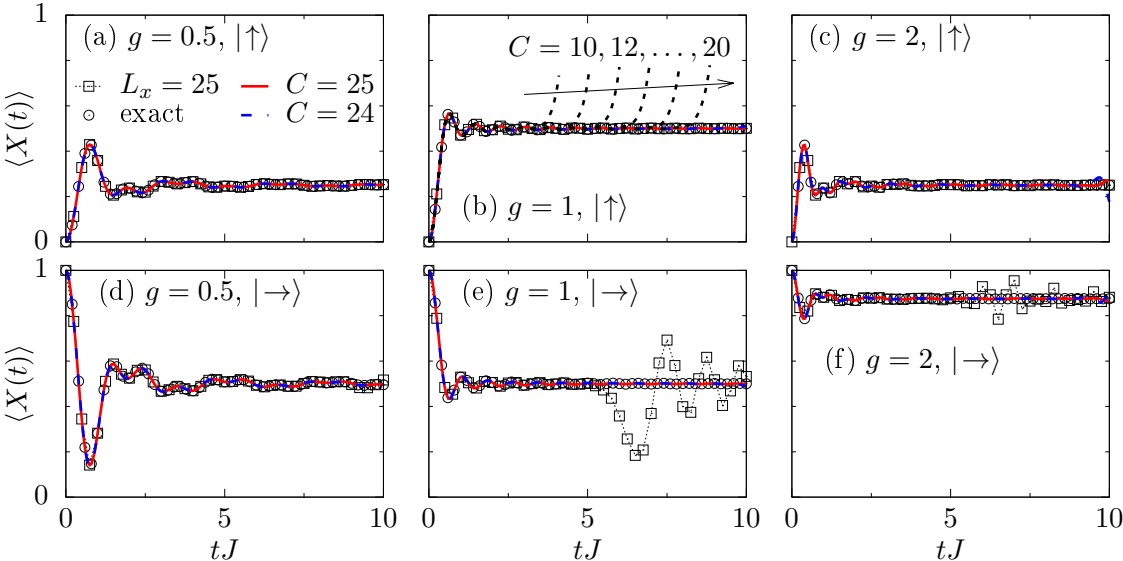

Figure 2: Dynamics of the transverse magnetization $\langle X(t) \rangle$ resulting from the initial state $|\psi(0)\rangle = |\uparrow\rangle$ [(a) - (c)], or $|\psi(0)\rangle = |\rightarrow\rangle$ [(d) - (f)], for chains with transverse fields $g = 0.5, 1, 2$. Numerical data obtained by NLCE for expansion orders $C = 24, 25$ (blue and red curves) are compared to direct simulations for chains with $L_x = 25$ and PBC (open boxes), as well as to the exact, analytically known result [86–88] given in Eq. (18) (open circles). In panel (b), we exemplarily show additional NLCE data for lower expansion orders $C = 10, 12, \ldots, 20$.

chain with $L_x = 25$ and PBC, and (ii) the exact, analytically known result [see Eq. (18) in Appendix A]. Note that we here choose to compare to systems with PBC, since finite-size effects are typically weaker in this case. For an additional comparison of NLCE results (also with lower expansion orders) to direct simulations of systems with OBC, see Appendix C.

Starting from its initial value $\langle X(0) \rangle = 0$, we find that the transverse magnetization $\langle X(t) \rangle$ in Figs. 2 (a)-(c) quickly increases and exhibits a peak at short times, before equilibrating towards a constant long-time value. This stationary value is reached already for times $tJ \approx 2$. While this overall behavior of $\langle X(t) \rangle$ is very similar for all values of $g$ considered, the long-time value $\langle X(t \rightarrow \infty) \rangle$ is found to vary with $g$. In particular, it is known that this long-time value can be described in terms of a suitable GGE [28].

Generally, we find that the NLCE results in Figs. 2 (a)-(c) are well converged on the time scales depicted, i.e., the curves for expansion orders $C = 24$ and $C = 25$ agree convincingly with each other. To visualize the convergence properties of the NLCE further, Fig. 2 (b) shows additional NLCE data for lower expansion orders $C = 10, 12, \ldots, 20$. Apparently, the convergence time of the expansion gradually increases with increasing $C$. Furthermore, we find that the curves for the finite chain with $L_x = 25$ also nicely coincide with the NLCE data for $L \rightarrow \infty$, i.e., finite-site effects appear to be less relevant in these cases. Importantly, our numerical results for $\langle X(t) \rangle$ agree perfectly with the analytical solution.

Next, in Figs. 2 (d)-(f), we consider quenches starting from the state $|\psi(0)\rangle = |\rightarrow\rangle$. Despite the obvious difference that $\langle X(t) \rangle$ now starts at a maximum, $\langle X(0) \rangle = 1$, the general picture is very similar compared to the previous case of $|\psi(0)\rangle = |\uparrow\rangle$. Namely, $\langle X(t) \rangle$ exhibits a rapid decay and equilibrates rather quickly towards its long-time value. Especially for $g = 1$ [Fig. 2 (e)], however, we now observe pronounced finite-size effects, i.e., the curve for $L_x = 25$ deviates from the analytical solution for times $tJ \gtrsim 5$ and exhibits oscillations. In contrast, the NLCE results for $C = 24, 25$ remain converged up to at least $tJ = 10$. This is a remarkable

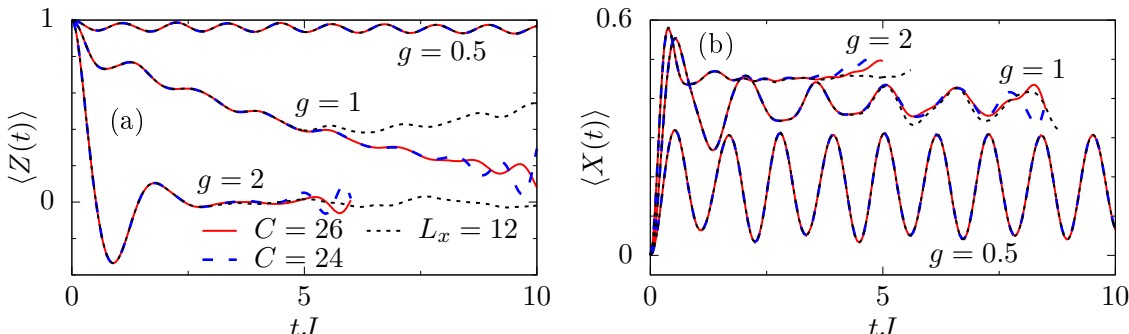

Figure 3: Dynamics of the (a) longitudinal magnetization $\langle Z(t)\rangle$ and (b) transverse magnetization $\langle X(t)\rangle$, in two-leg ladders with initial state $|\psi(0)\rangle = |\uparrow\rangle$ and different transverse fields $g$. Numerical data obtained by NLCE for expansion orders $C = 24, 26$ are compared to direct simulations for ladders with $L_x = 12$ and PBC.

result since the largest cluster in the NLCE also only has 25 lattice sites, i.e., the computational complexities of the NLCE and the simulation of the finite system are essentially the same.

Depending on the details of the quench, we thus find that performing a NLCE can yield a numerical advantage over the direct simulation of finite systems, see also Appendix C. On the one hand, if finite-size effects are weak, the results for finite chains can be very similar to the actual $L \to \infty$ dynamics (and also remain meaningful on longer time scales where the NLCE breaks down). On the other hand, the presence of strong finite-size effects [e.g. at the quantum critical point, cf. Fig. 2 (e)] appears to favor the usage of NLCEs which yield the dynamics directly in the thermodynamic limit. This is a first result of the present paper. As will be discussed in more detail in the upcoming sections, a similar parameter-dependent advantage (or disadvantage) of performing a NLCE occurs for ladder geometries and two-dimensional lattices as well.

## 4.2 Ladders

Let us now turn to the results for two- and three-leg ladders, which can be seen as intermediate cases between the chain geometry (cf. Sec. 4.1) and the two-dimensional square lattice (cf. Sec. 4.3). Since exact solutions for the dynamics of ladders are absent, we cannot compare our numerical data to analytical results. (For additional remarks on the transition from integrability to nonintegrability, see also Appendix D.)

In Fig. 3, we consider quenches starting from the state $|\psi(0)\rangle = |\uparrow\rangle$ in two-leg ladders with different transverse fields $g$. Here, the data is obtained by NLCE for expansion orders $C = 24$ and $C = 26$, i.e., the largest clusters involved are of size $12 \times 2$ or $13 \times 2$. As shown in Fig. 3 (a), the dynamics of the longitudinal magnetization $\langle Z(t)\rangle$ displays a strong dependence on the value of $g$. On the one hand, for $g = 2$, $\langle Z(t)\rangle$ rapidly decays, exhibits a minimum at $tJ \approx 1$, and equilibrates to zero for $tJ \gtrsim 3$. On the other hand, for $g = 1$, the decay of $\langle Z(t)\rangle$ towards zero is distinctly slower and much more monotonous. Moreover, for $g = 0.5$ (i.e. a quench within the same equilibrium phase), the decay of $\langle Z(t)\rangle$ is almost indiscernible on the time scale shown, and we additionally observe that $\langle Z(t)\rangle$ exhibits small oscillations for this value of $g$. The corresponding dynamics of the transverse magnetization $\langle X(t)\rangle$ is shown in Fig. 3 (b). While $\langle X(t)\rangle$ quickly equilibrates towards a stationary value for $g = 2$, $\langle X(t)\rangle$ displays oscillations for $g = 0.5, 1$ which, especially in the case of $g = 0.5$, do not equilibrate on the time scale shown here.

Let us comment on the convergence properties of the NLCE data in Fig. 3. Both for $\langle Z(t)\rangle$ and $\langle X(t)\rangle$, we observe that the NLCE remains converged for longer times if the value of $g$ is

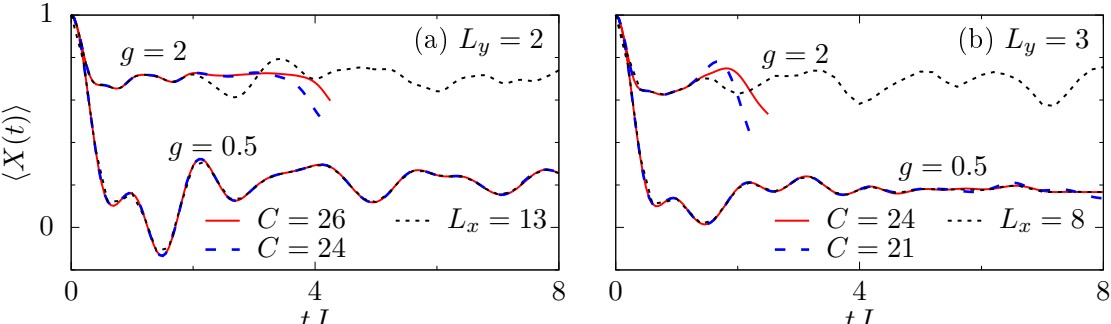

Figure 4: Dynamics of the transverse magnetization $\langle X(t) \rangle$ resulting from the initial state $|\psi(0)\rangle = |\rightarrow\rangle$ for (a) two-leg ladders and (b) three-leg ladders with $g = 0.5$ and $g = 2$. Numerical data obtained by NLCE for different expansion orders $C$ are compared to direct simulations of finite ladders with PBC. For additional NLCE data with lower expansion orders and a comparison to direct simulations of systems with OBC, see Appendix C.

smaller. Specifically, we find that the series breaks down at $tJ \approx 4$ for $g = 2$, at $tJ \approx 8$ for $g = 1$, while no breakdown can be seen for $g = 0.5$. Comparing these NLCE data to direct simulations of ladders with periodic boundary conditions and $L_x = 12$, a good agreement is found on short to intermediate time scales (or even longer for $g = 0.5$). In particular, the simulation for the finite ladder turns out to be advantageous for a strong transverse field $g = 2$, since it captures the stationary value of $\langle Z(t) \rangle$ and $\langle X(t) \rangle$ for a longer time than the NLCE. Similar to our previous results for chains, however, it becomes clear from Fig. 3 (a) that the usage of NLCEs is in turn beneficial for $g = 1$, where finite-size effects appear to be stronger and the NLCE captures the monotonous decay of $\langle Z(t) \rangle$ up to longer times compared to the finite-system data.

To proceed, Fig. 4 shows results for quantum quenches starting from the initial state $|\psi(0)\rangle = |\rightarrow\rangle$, with data for two-leg ladders in Fig. 4 (a) and data for three-leg ladders in Fig. 4 (b). Since $\langle Z(t) \rangle = 0$ due to the spin-flip symmetry of $\mathcal{H}$, we only have to consider $\langle X(t) \rangle$ in this case. We find that $\langle X(t) \rangle$ generally behaves very similar for the two different ladder geometries. Specifically, $\langle X(t) \rangle$ rapidly decays towards an (approximately constant) stationary value which is naturally higher for a higher value of $g$. Note however, that for $L_y = 2$ and $g = 0.5$, as well as for $L_y = 3$ and $g = 2$, $\langle X(t) \rangle$ still exhibits some residual fluctuations, i.e., perfect equilibration is absent. Concerning the convergence properties of the NLCE, we find that analogous to the previous case of $|\psi(0)\rangle = |\uparrow\rangle$ (cf. Fig. 3), the NLCE remains converged significantly longer for $g = 0.5$ compared to $g = 2$. Especially the early breakdown of convergence for $L_y = 3$ and $g = 2$ in Fig. 4 (b) emphasizes the fact that NLCEs are not necessarily the method of choice if one aims to study thermalization which typically requires the analysis of long time scales. (There exist, however, also examples where NLCEs yield converged results even in the infinite-time limit, i.e., converged results for the so-called diagonal ensemble [70].) Eventually, let us note that the NLCE results and the data for systems with PBC in Fig. 4 yield a considerably longer convergence compared to analogous simulations for systems with OBC, see Appendix C for details.

As a side remark to conclude the study of ladder geometries, let us note that Ref. [97] has recently discussed the possibility of quantum scars in transverse-field Ising ladders. Specifically, Ref. [97] has considered small values of $g$ and "density-wave" initial states of the form $|\psi(0)\rangle \sim |\uparrow\downarrow\uparrow\downarrow \cdots\rangle$. These initial states were found to exhibit a large overlap with rare, weakly entangled eigenstates, leading to quasi-periodic revivals in the dynamics. As detailed in Appendix D, the fully polarized states $|\uparrow\rangle$ and $|\rightarrow\rangle$ studied in the present paper, in contrast, do

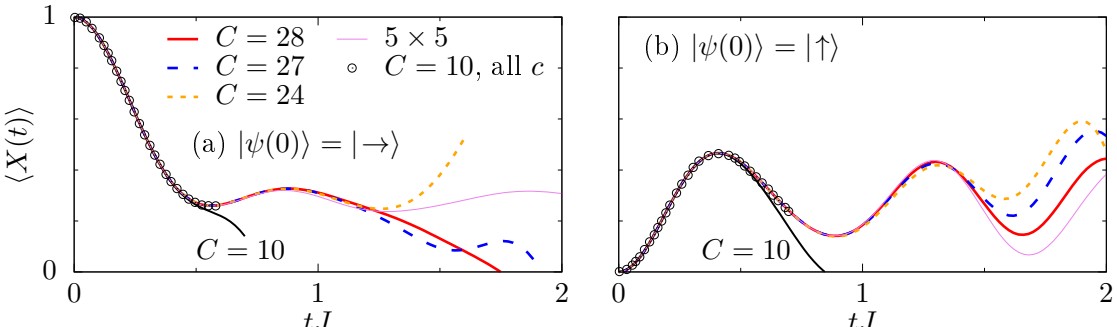

Figure 5: Dynamics of the transverse magnetization $\langle X(t) \rangle$ for a two-dimensional square lattice with transverse field $g = 1$, obtained by NLCE with rectangular clusters and expansion orders $C = 24, 27, 28$. The open circles are NLCE data digitized from Ref. [72], where all (also nonrectangular) cluster geometries with up to 10 lattice sites have been considered. In addition, we present data from a rectangle expansion up to $C = 10$ which, in comparison, converges to slightly shorter times than the full expansion. The dynamics for a $5 \times 5$ lattice with PBC is shown as well. The initial state is chosen as (a) $|\psi(0)\rangle = |\rightarrow\rangle$ and (b) $|\psi(0)\rangle = |\uparrow\rangle$.

not exhibit such a significant overlap with the weakly entangled eigenstates. These special eigenstates therefore do not play a distinguished role for the quench dynamics presented in Figs. 3 and 4.

## 4.3 Two-dimensional square lattice

We now come to the last part of this paper, i.e., the quantum quench dynamics in the two-dimensional transverse-field Ising model. Note that dynamical properties of this model [50, 52, 56, 58, 98, 99], as well as the emergence of thermalization [24, 25, 61], have been studied before by a variety of approaches. By comparing our results to existing data from the literature, let us demonstrate in this section that numerical linked cluster expansions based on rectangular clusters only, combined with an efficient forward propagation of pure states, provide a competitive alternative to other state-of-the-art numerical approaches.

As a first step, it is instructive to compare our results to earlier NLCE data from Ref. [72]. This comparison is shown in Figs. 5 (a) and (b), where the dynamics of the transverse magnetization $\langle X(t) \rangle$ is studied for quenches from $|\rightarrow\rangle$ and $|\uparrow\rangle$ with $g = 1$. (Recall that $g_c \approx 3.044$ for the two-dimensional lattice.) Importantly, Ref. [72] has considered all (also nonrectangular) cluster geometries in the expansion and has used full ED to evaluate the respective weights. Due to the computational bottlenecks of NLCEs discussed in Sec. 3.1, Ref. [72] was consequently limited to rather small clusters with up to 10 lattice sites. In Fig. 5, we find that our NLCE with solely rectangular clusters nicely reproduces the data from Ref. [72]. In particular, while the results of Ref. [72] are converged for times $tJ < 1$, the rectangular NLCE up to expansion order $C = 28$ (i.e. the largest clusters are of size $7 \times 4$, $14 \times 2$, $28 \times 1$) yields converged results on time scales which are approximately twice as long. This demonstration, that a NLCE restricted to rectangular cluster geometries can be better than a NLCE comprising all (possibly nonrectangular) clusters, is an important result of the present paper.

Let us add some comments on the convergence properties of the NLCE in Fig. 5. First, as an additional comparison between the rectangle expansion and the full expansion from Ref. [72], Figs. 5 (a) and (b) also show data obtained by the rectangle expansion with the lower expansion order $C = 10$. For this value of $C$, we find that the rectangle expansion is converged to slightly shorter times than the data from Ref. [72]. This is expected since, for a fixed value of

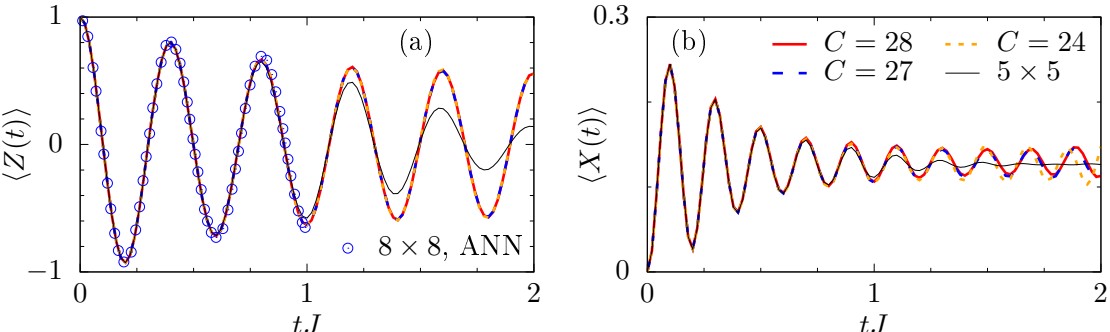

Figure 6: Dynamics of the (a) longitudinal magnetization $\langle Z(t)\rangle$ and (b) transverse magnetization $\langle X(t)\rangle$, resulting from the initial state $|\psi(0)\rangle = |\uparrow\rangle$ for a two-dimensional square lattice with $g = 2.63g_c$. Data obtained by NLCE for expansion orders $C = 24, 27, 28$ are compared to a simulation of a $5 \times 5$ lattice with PBC. In (a), we additionally show digitized ANN data from Ref. [56] for a $8 \times 8$ lattice.

$C$, the full expansion should always perform equally well or better compared to any restricted NLCE. However, let us stress once again the crucial advantage of the rectangle expansion that higher expansion orders can be included due to the reduced combinatorial costs. Secondly, we note that given the NLCE results up to expansion order $C = 28$ in Figs. 5 (a) and (b), the short-time dynamics for this value of the transverse field can apparently be accessed also by the direct simulation of a $5 \times 5$ lattice with PBC. This is similar to our previous findings for chains and ladders in Secs. 4.1 and 4.2. Namely, depending on the parameter regime, NLCEs might not necessarily outperform a direct simulation of a finite system with PBC if the latter yields small finite-size effects. As shown in Appendix C, however, the advantage of the NLCE is more pronounced when one compares to direct simulations of systems with OBC instead.

Next, let us study quenches starting from the state $|\psi(0)\rangle = |\uparrow\rangle$ such that $\langle Z(0)\rangle = 1$ and $\langle X(0)\rangle = 0$, and consider a strong transverse field $g = 2.63g_c \approx 8$, i.e., a quench across the quantum critical point. Again, we consider clusters with up to 28 lattice sites in the NLCE. In Fig. 6 (a), we find that $\langle Z(t)\rangle$ displays pronounced oscillations with an amplitude that is weakly damped over time. Correspondingly, the transverse magnetization $\langle X(t)\rangle$ in Fig. 6 (b) exhibits damped oscillations as well (with a frequency that is twice as large). It is instructive to compare these NLCE data for the thermodynamic limit to a simulation of a $5 \times 5$ lattice with PBC. Specifically, one observes that for such a finite system and times $tJ \gtrsim 1$, the oscillations of $\langle Z(t)\rangle$ and $\langle X(t)\rangle$ die away rather quickly. This is in contrast to the NLCE results for $L \to \infty$ which capture the persistent oscillations on a longer time scale. In addition, we compare our NLCE results for $\langle Z(t)\rangle$ in Fig. 6 (a) to recent data digitized from Ref. [56], which are computed by an artificial neural-network (ANN) approach for a $8 \times 8$ lattice. While the NLCE and ANN data agree nicely with each other for times $tJ < 1$, the NLCE remains converged also on longer time scales. In particular, the ANN data from Ref. [56] up to times $tJ \lesssim 1$ can be reproduced even by the smaller $5 \times 5$ lattice. Thus, for the parameter regime considered in Fig. 6, it appears that the NLCE can be better than the direct simulation of finite systems with PBC as well as the ANN approach from Ref. [56]. This is another important result of the present paper.

Finally, we also consider quenches starting from the state $|\psi(0)\rangle = |\rightarrow\rangle$. The values of the transverse field are chosen as $g = 0.1g_c, 1g_c, 2g_c$, which again allows us to compare to ANN data from Ref. [56], as well as to data from Ref. [52] based on infinite projected entangled pair states (iPEPS). For all values of $g$ shown in Figs. 7 (a)-(c), we find a convincing agreement between the data from Refs. [52, 56] and our NLCE results up to expansion order $C = 28$, with convergence times that are rather similar for all three methods. In order to put the convergence

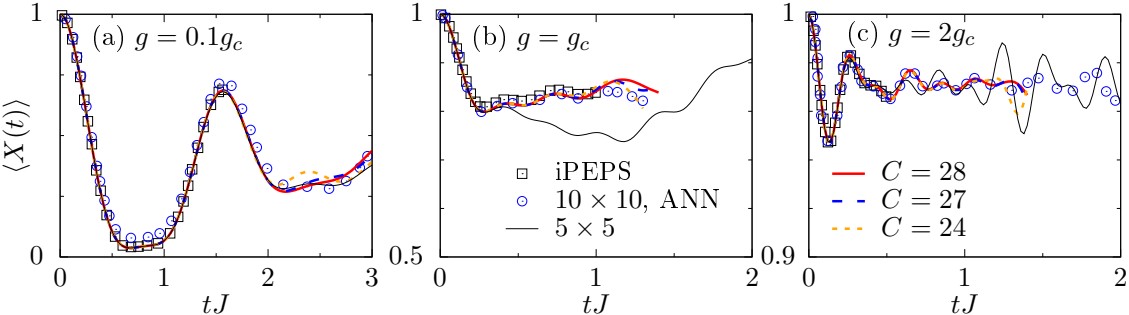

Figure 7: Dynamics of the transverse magnetization $\langle X(t) \rangle$ for two-dimensional lattices with initial state $|\psi(0)\rangle = |\rightarrow\rangle$ and transverse fields (a) $g = 0.1g_c$, (b) $g = 1g_c$, and (c) $g = 2g_c$. Data obtained by NLCE for expansion orders $C = 24, 27, 28$ are compared to the simulation of a $5 \times 5$ lattice with PBC. Additionally, we show iPEPS data digitized from Ref. [52] and ANN data for a $10 \times 10$ lattice digitized from Ref. [56].

times into perspective, it is again helpful to compare the NLCE data to a simulation of a finite $5 \times 5$ lattice with PBC. While finite-size effects appear to be less important for $g = 0.1g_c$ and $g = 2g_c$, we observe pronounced finite-size effects for $g = g_c$ already at short times $tJ \approx 0.5$ due to, e.g., the divergence of the relevant length scales at the quantum critical point. Importantly, the NLCE results for $g = g_c$ in Fig. 7 (b) remain converged up to times $tJ \approx 1.5$. One explanation for the advantage of NLCEs at the quantum critical point might be given by the fact that the expansion involves a variety of clusters with different ratios of width and height such that one can capture the dynamics on longer time and length scales. This is another central result of this paper. In this context, let us add that the inclusion of rectangles with different length ratios appears to be crucial to achieve a good convergence. For instance, we have checked that an expansion using solely square-shaped clusters $(1 \times 1, 2 \times 2, \ldots, 5 \times 5)$ performs very poorly instead (not shown here).

# 5 Conclusion

To summarize, we have studied the nonequilibrium dynamics of the transverse and the longitudinal magnetization resulting from quantum quenches with fully polarized initial states in the transverse-field Ising model defined on different lattice geometries. To this end, we have relied on an efficient combination of numerical linked cluster expansions and a forward propagation of pure states via Chebyshev polynomials.

Depending on the geometry and the parameter regime under consideration, the quench dynamics has been found to display a variety of different behaviors ranging from quick equilibration, over slower monotonous relaxation, to persistent (weakly damped) oscillations. As a main result, we have demonstrated that NLCEs comprising solely rectangular clusters provide a promising approach to study the dynamics of two-dimensional quantum many-body systems directly in the thermodynamic limit. While the organization of the NLCE becomes straightforward due to the simple cluster geometry, the memory efficient pure-state propagation made it possible to include clusters with up to 28 lattice sites. Especially, for quenches to the quantum critical point, where finite-size effects are typically strong, we have shown that NLCEs can yield converged results on time scales which compare favorably to direct simulations of finite systems with periodic boundary conditions (also in the case of chains or ladders). By comparing to existing data from the literature, we have demonstrated that the reachable time scales are also competitive to other state-of-the-art numerical methods. While NLCEs with rectangular

clusters have been used before to obtain thermodynamic quantities [100] or entanglement entropies [68], the present paper unveils that such NLCEs also provide a powerful tool to study the real-time dynamics of quantum many-body systems.

A natural direction of future research is to further explore the capabilities of NLCEs to simulate quantum quench dynamics of two-dimensional systems in the thermodynamic limit. In this context, it might be promising to consider other building blocks for the expansion such as, e.g., clusters that consist of multiple corner-sharing 2 × 2 squares [66]. Moreover, it will be interesting to study other two-dimensional lattice geometries such as triangular or Kagome lattices with nonrectangular cluster shapes. One particular question in the field of quantum many-body dynamics where NLCEs might be able to contribute is the existence of many-body localization in higher dimensions. While the usage of NLCEs in disordered systems involves additional complications beyond our explanations in Sec. 3.1, NLCEs can yield results directly in the thermodynamic limit which is especially important close to the potential transition between the thermal and the MBL regime. Although truly long times might still remain out of reach, the usage of supercomputing will be helpful to include higher expansion orders (up to $C \approx 40$ [76]), which improves the convergence of the NLCE even further.

*Note added:* After this paper was submitted, we became aware of the related work [101] which appeared in the same arXiv posting as our manuscript. While Ref. [101] also presents NLCE calculations for the dynamics of two-dimensional systems using an expansion in rectangles, its focus is on the application of NLCEs to disordered systems and inhomogeneous initial states. In addition, while Ref. [101] employs full ED to evaluate the contributions of the clusters, the present paper highlights the usefulness of efficient pure-state propagation methods to reach expansion orders beyond the range of full ED and to extend the convergence times of the NLCE.

# Acknowledgements

The authors thank F. Jin for very helpful discussions.

**Funding information** This work has been funded by the Deutsche Forschungsgemeinschaft (DFG) - Grants No. 397067869 (STE 2243/3-1), No. 355031190 - within the DFG Research Unit FOR 2692.

# A Exact solution for the integrable chain

In the case of a chain geometry, the transverse-field Ising model (1) is a paradigmatic example of an integrable model and can be diagonalized by means of subsequent Jordan-Wigner, Fourier, and Bogolioubov transforms [60],

$$\mathcal{H} = \sum_k E_k \eta_k^\dagger \eta_k + \text{const.}, \quad E_k = 2J\sqrt{(g - \cos k)^2 + \sin^2 k}. \tag{17}$$

Since quantum quenches in the transverse-field Ising chain have been studied extensively before, and since the focus of this paper is on the numerical analysis of nonintegrable geometries, we here refrain from providing more details and refer to the large body of existing literature instead [89–96]. Given the notation of $\mathcal{H}$ in Eqs. (1) and (17), as well as an initial state $|\psi(0)\rangle$ which is chosen as the groundstate of $\mathcal{H}$ for some transverse field $g'$, the dynamics of

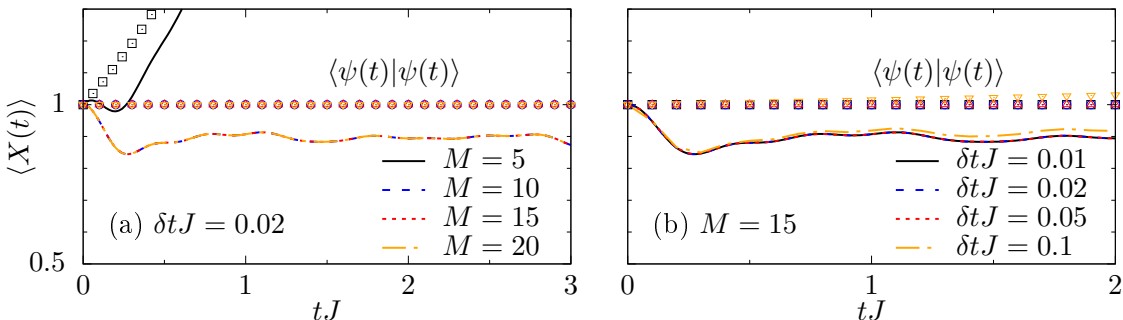

Figure 8: Dynamics of the transverse magnetization $\langle X(t)\rangle$ for a cluster of size $L_x \times L_y = 7 \times 3$ (with OBC), initial state $|\psi(0)\rangle = |\rightarrow\rangle$, and transverse field $g = 3.044$. The symbols indicate the norm $\langle\psi(t)|\psi(t)\rangle$. (a) Fixed time step $\delta t J = 0.02$ and varying expansion order $M$. (b) Fixed $M = 15$ and varying $\delta t J$.

the transverse magnetization $\langle X(t)\rangle$ for a quench $g' \rightarrow g$ is then given by [86–88, 90],

$$\langle X(t)\rangle = 2 \int_0^\pi \frac{dk}{2\pi} \frac{1}{E_k^2 E_k'} \left[ \epsilon_k(\epsilon_k \epsilon_k' + \gamma_k^2) + \gamma_k^2(\epsilon_k' - \epsilon_k)\cos(2E_k t) \right] , \qquad (18)$$

where we have used the abbreviations

$$\epsilon_k = 2J(g - \cos k) , \quad \gamma_k = 2J \sin k , \qquad (19)$$

and $E_k'$ and $\epsilon_k'$ are defined like their unprimed counterparts, but with $g \rightarrow g'$. In order to obtain the results shown in Fig. 2 of the main text, we have numerically evaluated the integral in Eq. (18) either for $g' = 0$ ($|\psi(0)\rangle = |\uparrow\rangle$) or for $g' \rightarrow \infty$ ($|\psi(0)\rangle = |\rightarrow\rangle$).

# B Accuracy of the pure-state propagation

While we have already demonstrated that our numerical results agree very well with existing data, let us nevertheless discuss the accuracy of the Chebyshev-polynomial expansion which is used to evaluate the time evolution of the pure states $|\uparrow\rangle$ and $|\rightarrow\rangle$. To this end, Fig. 8, shows the dynamics of the transverse magnetization $\langle X(t)\rangle$ for a cluster of size $L_x \times L_y = 7 \times 3$ (with OBC), initial state $|\psi(0)\rangle = |\rightarrow\rangle$, and transverse field $g = g_c \approx 3.044$.

First, in Fig. 8 (a), we set the discrete time step to $\delta t J = 0.02$ and depict results for different expansion orders $M = 5, 10, 15, 20$ (curves). On the one hand, for small $M = 5$, we observe clearly unphysical results (e.g. $\langle X(t)\rangle > 1$), which can also be explained by the fact that the norm $\langle\psi(t)|\psi(t)\rangle$ (symbols) is not conserved over time for this choice of $M$. On the other hand, for $M = 10, 15, 20$, all curves for $\langle X(t)\rangle$ are perfectly on top of each other, i.e., convergence with respect to $M$ has been reached, and $\langle\psi(t)|\psi(t)\rangle = 1$.

Next, Fig. 8 (b) shows results for a fixed expansion order $M = 15$ and varying time step $\delta t J = 0.01, 0.02, 0.05, 0.1$. We find that $\langle X(t)\rangle$ is practically independent of the time step for the three smallest values of $\delta t J$ used here. However, visible deviations occur in the case of the largest time step $\delta t J = 0.1$. While the required time step $\delta t$ and expansion order $M$ can certainly depend on the parameter regime under consideration, the typical choice used in the main text, i.e., $\delta t J = 0.02$ and $M = 15$, appears to yield very accurate results.

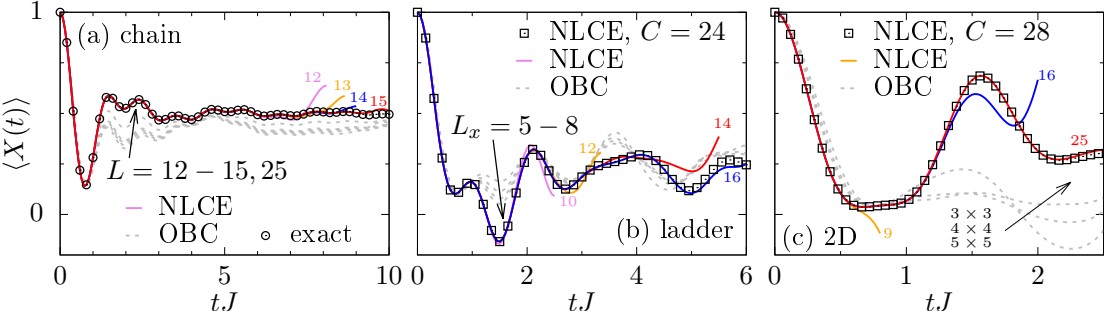

Figure 9: Dynamics of the tranverse magnetization $\langle X(t) \rangle$ for quenches in (a) chains with $g = 0.5$, (b) two-leg laddders ($L = L_x \times 2$) with $g = 0.5$, and (c) two-dimensional lattices with $g = 0.1g_c$. The initial state is $|\psi(0)\rangle = |\rightarrow\rangle$. Results obtained by numerical linked cluster expansion for different expansion orders $C$ (solid curves) are compared to direct simulations for systems with open boundary conditions (gray dashed curves). In all cases, we find that for a given expansion order $C$ (or system size $L$) the NLCE yields converged results for significantly longer times than the corresponding direct simulation with OBC.

## C  Additional results for lower expansion orders and systems with open boundary conditions

In Figs. 2 - 7 of the main text, we have compared the convergence of the NLCE to direct simulations of systems with periodic boundary conditions. However, since the clusters entering the NLCE are defined with open boundary conditions, it might be interesting to compare the convergence of the NLCE to direct simulations with OBC as well. Such a comparison is shown in Fig. 9 for the dynamics of the transverse magnetization $\langle X(t) \rangle$ in chains [panel (a)], ladders [panel (b)], and two-dimensional square lattices [panel (c)], with one exemplarily chosen transverse field $g$ in each case. Specifically, the curves shown in Fig. 9 are complementary to our earlier data in Figs. 2 (d), 4 (a), and 7 (a), as we now also include NLCE results for lower expansion orders. Moreover, to guarantee a fair comparison, Fig. 9 always shows curves for matching system sizes and expansion orders, i.e., $L = C$ (recall that the expansion order of the NLCE is defined as the largest cluster size involved in the expansion). Importantly, we find that the NLCE yields converged results on significantly longer time scales than the simulation of the finite system with OBC for all cases shown here. For instance, in the case of the chain [Fig. 9 (a)], expansion order $C = 15$ is already sufficient to yield converged results up to $tJ = 10$, whereas the direct simulation for a system with OBC fails to capture the correct long-time plateau even for the considerably larger system size $L = 25$. Similarly, in the case of the two-leg ladder [Fig. 9 (b)], the curves for finite systems with OBC converge only up to the rather short time $tJ \approx 1$, while the convergence of the NLCE quickly improves with increasing $C$. Especially for the two-dimensional case [Fig. 9 (c)], the simulations for the finite system with OBC even fail to describe the initial decay of $\langle X(t) \rangle$ correctly. Thus, we conclude that for a given expansion order $C$ (or system size $L$) the NLCE performs considerably better than the corresponding direct simulation with OBC.

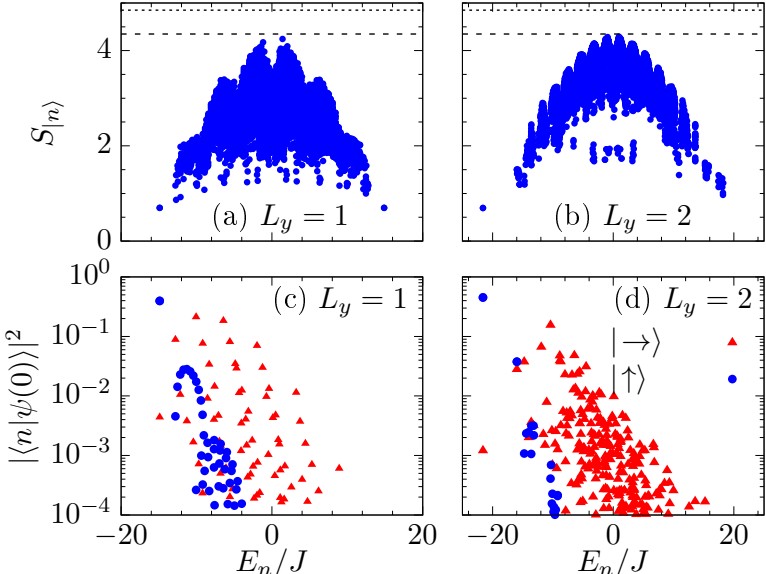

Figure 10: (a) and (b) Eigenstate entanglement entropy $S_{|n\rangle}$ of a chain and a two-leg ladder. The dashed line indicates the "Page value" for a random state [106], while the dotted line indicates the maximum entropy possible for the chosen bipartition. (c) and (d) Overlap of initial states $|\uparrow\rangle$ and $|\rightarrow\rangle$ with the eigenstates $|n\rangle$. We have $L = 14$, $g = 0.5$, and PBC in all cases.

## D Eigenstate entanglement and spectral decomposition of initial states

Let us discuss some properties of the fully polarized initial states $|\uparrow\rangle$ and $|\rightarrow\rangle$. To this end, we first study the entanglement (von Neumann) entropy $S_{|n\rangle}$ of the eigenstates $|n\rangle$ of $\mathcal{H}$,

$$S_{|n\rangle} = -\text{Tr}[\rho_A \ln \rho_A], \quad \rho_A = \text{Tr}_B\{|n\rangle\langle n|\}, \tag{20}$$

where $\rho_A$ is the reduced density matrix on a subsystem $A$, obtained by tracing over the degrees of freedom in the complement $B$. In Figs. 10 (a) and (b), $S_{|n\rangle}$ is shown for a chain and a two-leg ladder respectively, numerically obtained by full exact diagonalization for $L = 14$ sites, transverse field $g = 0.5$, and periodic boundary conditions. In both cases, we have chosen $A$ as one half of the system, i.e., the first 7 lattice sites in case of the chain, or the first three rungs and one site of the fourth rung in case of the ladder. On the one hand, for the integrable chain geometry in Fig. 10 (a), we find that $S_{|n\rangle}$ is comparatively small at the edges (consistent with the area-law entanglement scaling of groundstates [102]), while weakly and strongly entangled states coexist in the bulk of the spectrum (see also Refs. [103, 104]). On the other hand, for the two-leg ladder in Fig. 10 (b), the fluctuations of $S_{|n\rangle}$ in the center of the spectrum are clearly smaller, i.e., the eigenstates are typically stronger entangled. This behavior of $S_{|n\rangle}$ can be interpreted as an indication of the transition from integrability to nonintegrability [104], by going from chains to ladders. In addition, we can identify a small number of eigenstates $|n\rangle$ with energy close to $E = 0$ in Fig. 10 (b), which exhibit a distinctly lower value of $S_{|n\rangle}$. This appears to be consistent with the recent proposal of quantum scars in transverse-field Ising ladders in Ref. [97].

Next, it is useful to study $S_{|n\rangle}$ in combination with the overlap between the initial states

$|\psi(0)\rangle = |\uparrow\rangle, |\rightarrow\rangle$ and the eigenstates $|n\rangle$,

$$P_{|\psi\rangle} = \sum_{n=1}^{D} |\langle n|\psi(0)\rangle|^2 \delta(E - E_n) , \qquad (21)$$

where $E_n$ is the eigenvalue of $\mathcal{H}$ belonging to $|n\rangle$. As shown in Figs. 10 (c) and (d), this spectral distribution is narrow and peaked at the groundstate in the case of $|\uparrow\rangle$, while $P_{|\psi\rangle}$ is much broader for $|\rightarrow\rangle$, both for the chain and the ladder. Thus, a quench to $g = 0.5$ with $|\psi(0)\rangle = |\uparrow\rangle$, results in a dynamics which is strongly dominated by the groundstate with a significantly smaller admixture of excited states. Note that exactly for such a situation, i.e., a quantum many-body system with one macroscopically populated eigenstate, an analytical prediction for the temporal relaxation has been recently obtained in Ref. [105]. While this is beyond the scope of the present manuscript, it appears that quantum quenches in transverse-field Ising chains or ladders can be promising candidates to test such predictions.

Finally, as already pointed out in the main text, we note that the fully polarized initial states $|\uparrow\rangle$ and $|\rightarrow\rangle$ do not exhibit a distinguished overlap with the rare, weakly entangled eigenstates discussed in Fig. 10 (b). These potential quantum scars therefore do not play an essential role for the resulting quench dynamics.

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
