# Peer review of "Quantum quench dynamics in the transverse-field Ising model: A numerical expansion in linked rectangular clusters"

_SciPost Physics, doi:SciPost Phys. 9, 031 (2020)_

## Round 2 · Referee Report · Anonymous (Referee 1) · 2020-5-20

Strengths

1) propose method to simulate quantum quenches
2) demonstrate that the method is competitive to other state-of-the-art methods
2) well-written, strengths and weaknesses are properly addressed

Weaknesses

minor: missing some motivation for the method
minor: missing some motivation for particular choices in the algorithm

Report

The authors propose to combine numerical linked cluster expansion with sparse time evolution algorithms to study the dynamics of quantum quenches. The algorithm is tested on a transverse-field Ising model in various geometries and compared to results from different methods including an exact solution for chains. For the two-dimensional square lattice limit, their method is compared to other recent methods and is demonstrated to be competitive.

The manuscript is well-written and the results are worked out in proper detail. The authors properly highlight the strengths and weaknesses of the method and the presentation is scientifically sound. Their results demonstrate that the proposed method is competitive with other state-of-the-art methods, including Artificial neural networks and iPEPS. There are, however, some questions I would like to address before the publication of the manuscript.

1) The study of quantum quenches is a broad and interesting research field. I would like the authors to elaborate more on the potential applications of the proposed method. What specific questions can this method answer, given current limitations in cluster size? Why are those questions relevant? Could the times that can be reached using their method be long enough to gain insights into certain physical phenomena? It would be good to give more explicit motivation here and realistic estimation of which scientific problems might be tackled.

2) I would like the authors to comment on the choice of rectangular subclusters in the numerical linked cluster expansion. I guess the choice is motivated by having to compute observables on fewer clusters than if arbitrary subclusters are considered. How does this change the convergence of the series in the linked-cluster expansion? Does one have to simulate larger clusters to reach the same accuracy than if all arbitrarily shaped subclusters are considered? There are some remarks in the section about the two-dimensional limit about this, but I would like the authors to elaborate on this a bit more.

3) The authors choose a Chebyshev polynomial approach to perform the time evolution. It would be good to have motivation for this choice. Does the Chebyshev polynomial approach have favorable properties in this context as opposed to other methods like Lanczos time evolution, or Runge-Kutta methods?

4) In Figure 2 it is not clear what the black boxes and circles are. It is explained in the text, but putting an explanation in the Figure caption would improve readability.

5) Figures 2, 3, 4 show how well the NLCE works when going to a large order of the cluster expansion. It would be interesting to see which times can be reached by which order of the expansion. Therefore, it would help if also a few results at lower orders are shown. Since this should now be new calculations, it would be great to add this data.

6) The authors mention that "exact diagonalization" is limiting the system size that allows for time evolution and therefore use the sparse Chebyshev algorithm to do so. I think that one could be more specific here, and call this "full exact diagonalization". Applying sparse algorithms such as Lanczos or Chebyshev algorithms on full wave functions is also often called "exact diagonalization", so this could be confusing. This is, of course, a very minor remark.

I think the manuscript would be of interest to many readers and the method will be useful in future studies. Upon addressing the remarks above, I can recommend the publication in SciPost.

Requested changes

see report

  • validity: top
  • significance: high
  • originality: good
  • clarity: top
  • formatting: excellent
  • grammar: excellent

Author:  Jonas Richter  on 2020-07-24  [id 905]

(in reply to Report 1 on 2020-05-20)

We thank this Referee very much for writing a report on our manuscript, as well as for her/his overall positive assessment of our work. Our response is structured according to her/his report.

1) As we have already mentioned in the introduction, one major motivation is given by quantum simulator experiments on two-dimensional lattices. On the one hand, unbiased numerical methods (such as NLCE) are important to confirm the accuracy of such experiments. On the other hand, numerical results might also serve as an orientation for experiments to explore the dynamics for certain models or parameter regimes in more detail. One particular question in the field of quantum many-body dynamics, where we hope that NLCEs can contribute,
is the existence (or absence) of many-body localization in higher dimensions. Although the usage of NLCEs in disordered systems involves additional complications (e.g. translations of the cluster are not equivalent anymore due to the disorder), the availability of results in the thermodynamic limit can be important especially close to the potential transition between the thermal and the MBL regime.

Certainly, the reachable time scales using NLCEs appear to be limited (although we show in the manuscript that the time scales can be distinctly longer depending on the parameter regime). Let us note, however, that other available numerical methods, such as the ones we compare to in this manuscript, likewise face this challenge of reaching longer times. Using supercomputing, the reachable cluster sizes within the NLCE can nowadays be pushed to expansions orders C \approx 40, which hopefully then gives rise to converged results on intermediate time scales which are long enough to extract meaningful results. In the revised manuscript, we have added some comments in the introduction as well as in the conclusion.

2) Indeed, the choice of rectangular clusters is motivated by the fact that one has to evaluate significantly less clusters and, as a consequence, can therefore reach larger expansion orders. A priori, it is however difficult to estimate how fast the series converges for this restricted set of clusters. If one fixes the expansion order, one would certainly expect that an expansion including all (arbitrarily shaped) clusters yields better results than an expansion including only rectangles. In the revised manuscript, we include new data in Fig. 5, where we directly compare the rectangle expansion and the full expansion for order C = 10. As expected, the full expansion performs slightly better in this case. However, while C = 10 is roughly the maximum for the full expansion due to combinatorial constraints, the rectangle expansion can reach larger expansion orders (here C = 28) and yields converged results on longer time scales. We have extended the corresponding discussion in the text.

3) Within the NLCE, it is crucial that the contribution of each cluster is evaluated with high accuracy. Even relatively small numerical errors for each individual cluster could potentially spoil the convergence of the series when the contributions are added together. In this context, the Chebyshev polynomial expansion can yield results essentially up to machine precision if its order and the time step are chosen appropriately.

Comparing the Chebyshev expansion to other possible methods, we believe that a similar accuracy can be achieved by a Lanczos time evolution as well, with roughly the same computational costs. A slight advantage of the Chebyshev expansion might be its lower memory consumption, as the Lanczos method typically requires the storage of the full Krylov basis in memory. Overall, however, a Lanczos time evolution would probably be a very good choice in the context of the present paper as well. Concerning Runge-Kutta methods, we have experienced that they can become rather unstable if the operator norm of the Hamiltonian is large (for the model studied here, this happens for instance if the transverse field is strong). In turn, the discrete time step of the Runge-Kutta method has to be chosen very small (significantly smaller compared to Chebyshev or Lanczos schemes) in order to yield reliable results. We have added some clarifying comments in the text.

4) In fact, there already was a key in Fig. 2(b) explaining the meaning of boxes and circles. In the revised version, we have now modified the positioning of the key and also slightly modified the caption.

5) We thank the Referee for this comment. We absolutely agree that the reader would benefit if results for lower expansion orders are shown as well. At the same time, we are also a bit worried that the figures become too overloaded in this case. As a consequence, we have decided to add a new section in the Appendix where we exemplarily show corresponding data for lower expansion orders and also for systems with open boundary conditions. Moreover, we have added some data for lower expansion orders in Fig. 2(b) and Fig. 5 in the main text. We have added corresponding comments in the text.

6) We thank the Referee for this suggestion. We have changed the text accordingly.

We once again would like to thank the Referee for her/his comments which have helped us to improve our manuscript. Given our response and the changes made to the text, we hope that our manuscript is now ready for publication in SciPost Physics.

---

## Round 2 · Referee Report · Anonymous (Referee 2) · 2020-6-4

Strengths

1-Potentially interesting method development for quantum dynamics in higher dimensions
2-Clearly written manuscript

Weaknesses

1-The actual advantage of the method is not fully clear
2-Convergence of the cluster expansion does not seem to be easily controlled

Report

The manuscript explores quench dynamics in the transverse field Ising model in a chain, ladders and 2D latices using a combination of linked cluster expansion and Chebyshev polynomial expansion for pure state evolution. The paper focuses on method development, the physics explored in the paper has been studied with other methods and therefore is used for comparison. Time evolution with linked cluster expansion has been studied before, even in this same model. The two new aspects of the method introduced in this paper is: i) selecting a subset of all possible clusters in the linked cluster expansion, and ii) using methods for the time evolution of a cluster that are more efficient than exact diagonalization, here they use the Chebyshev polynomial method.

The method the authors discuss is in principle interesting. The authors argue that some of their results show that one can go beyond exact diagonalization and obtain results at least comparable to other state-of-the-art methods for time evolution in 2D. While I think this is essentially true, I think the data shown in the paper suggest that the method is maybe not much more powerful than direct evaluation. Some aspects of the discussion are not fully clear as expanded on below.

I have a few direct comments on the manuscript and calculations:

1-On page 4 the authors write: "... the exact value of $g_c$ can vary due to finite-size corrections". Though I think I understand what the authors mean, the critical field by definition can not depend on finite-size corrections. Maybe the authors can clarify.

2-In Eq. (6) the authors introduce the multiplicities $\mathcal{L}_c$, but don't really define it. It would be useful for the reader if this was defined carefully.

3-Eq. (6) sums over "connected" clusters. While one may intuitively guess what is meant by connected here it would be beneficial to define this. In particular, I guess this depends on the Hamiltonian. For example, on the square lattice, connected sites are only nearest neighbours and not for example sites $(i,j)$ and $(i+1,j+1)$, right?

4-In several places the authors discuss topologically distinct clusters. I am confused about what they are talking about. As far as I can see, all clusters that the authors study in this paper are simply connected clusters in the plane. Therefore, in fact all clusters that they discuss are topologically equivalent. I did not see any representation of topologically distinct clusters (which would then have to be not simply connected). Maybe they mean something else with this phrase? This needs to be clarified.

5-On page 5, the authors write that "Eq. (6) can also converge for different types of expansions." This is an essential point of the paper. They have considered one type of expansion. Do the authors have any idea of what the principle for identifying useful approximate expansion is? Could they explain this? Or does one need to just work with trial and error?

6-Right after introducing the different expansion I mentioned in point 5, the authors mention the cluster with rectangular shape. But from what follows, it is only a restricted set of rectangular shapes that are used, at least in the ladders. Maybe the authors could clarify this in the text.

7-In Fig. 1 and below Eq. (8), the authors note that a cluster c and the same cluster rotated by 90 degrees, are topologically equivalent and therefore enter the sum (6) with multiplicity 2. It's not clear, partially since the multiplicity was not carefully defined, what this means. Do they simply mean that they can simply restrict the sum and multiply with two instead of summing over both types. I'm not sure I would say that this is equivalent to having multiplicity 2 in Eq. (6), since that equation would also have to be restricted for this to be true. Also, since this is in a section that is discussing the method in general, I would think that these two clusters are only equivalent if the Hamiltonian is actually invariant under rotations by 90 degrees. Is this true? Also here in this discussion, the notion of topologically equivalent clusters is a bit confusing, since all the clusters in Fig. 1 are topologically equivalent (not only the two green ones).

8-At the bottom of Eq. (7) the authors list the symmetries that are not present and could therefore not be used to speed up the calculation. But there is a parity symmetry that could be used. Why do they not mention this?

9-Moving on to results. In Fig. 2 the authors compare their method to time evolution on a finite system with PBC. At the same time, the clusters used in the linked cluster expansion have OBC. In principle these will have different finite size corrections, and in particular have different recurrence times. It would perhaps make more sense to compare with simulation on finite samples with OBC? Would it be possible to add this data? Also, the authors only show data for the linked cluster expansion with C = 24 and C = 25, that is, with clusters that are equally large as the finite size simulation. It would be interesting, to understand better how finite cluster effects appear, to also see data with cluster expansion with smaller C. Together these two, finite size with OBC and clusters with C < L, would better help make the point of the authors that "NLCE can yield a numerical advantage over the direct simulation of finite systems," which I don't think is fully clear from this data.

10-In the ladder simulation, why us there no data for g = 1 in Fig. 3b? Here, also, it would be interesting to see that finite system data with OBC.

11-In Fig. 5, it seems that the direct simulation of a 5x5 cluster is essentially equally good as the NLCE. Does that mean there is no advantage to the NLCE? Maybe a bit more discussion here would be useful.

12-In the comparison with ANN, it seems that the ANN data was stopped in time while still being accurate. Does that mean that the ANN could actually go to larger times and be comparable with the NLCE?

13-Do the authors have a way of knowing if the subset of clusters that they choose in their cluster expansion converges to a reliable result? One could for example imagine that within this subset the series Eq. (6) would converge and therefore only comparing different C within this subset would indicate convergence, but that then this convergence is to a value that is not the same as the full sum.

Requested changes

The requested changes are listed in the relevant points above.

  • validity: high
  • significance: ok
  • originality: good
  • clarity: high
  • formatting: excellent
  • grammar: excellent

Author:  Jonas Richter  on 2020-07-24  [id 906]

(in reply to Report 2 on 2020-06-04)

We thank this Referee very much for writing a report on our manuscript, as well as for her/his overall positive assessment of our work. Our response is structured according to her/his report.

1) We thank the Referee for this comment. We hope that our phrasing in the revised version is now more precise.

2) In the revised version, we now provide a more detailed definition of the multiplicity ${\cal L}_c$.

3) The Referee is correct that for a two-dimensional square lattice and the definition of the Hamiltonian in Eq. (1), the sites (i,j) and (i+1,j+1) are not directly connected. Nevertheless, both sites can be part of a connected cluster which would then also include, e.g., the site (i+1,j). The notion of "connected cluster" refers to the fact that a cluster cannot be split into two "disconnected" parts (e.g. a couple of lattice sites in one region of the lattice and then a few other lattice sites in a different part of the lattice, without any terms in the Hamiltonian coupling these parts). In the revised manuscript, we have clarified the notion of a "connected cluster" and included the example provided by the Referee.

4) We agree that the notion of "topologically distinct" clusters is somewhat confusing (although we note that it is also used in review papers to NLCEs). What we actually want to say is that clusters which can be mapped onto each other by rotations or translations are topologically equivalent in the sense that these clusters yield the same contribution and only need to be calculated once [see also reply to point 7) below]. In the revised manuscript, we now avoid this potentially ambiguous wording.

5) As written in the text, a crucial bottleneck of NLCEs is the quickly growing number of different clusters which need to be evaluated. In this context, a first criterion for an approximate expansion to be "useful" is its simplicity. For the two-dimensional square lattice, the probably simplest choice is to use only square-shaped clusters, $1 \times 1$, $2 \times 2$, ..., $5 \times 5$ (the next largest $6\times 6$ cluster is already numerically unfeasible in many cases). However, we have tried this square expansion, and it turns out to yield very bad convergence times.

The rectangular expansion used in the present paper is also comparatively simple in the sense that only a small number of clusters needs to be evaluated and the organization of subclusters is straightforward as well. However, we believe that an important point of the rectangle expansion is the fact that it contains clusters with very different length scales, i.e., both square-like clusters but also ladder-like clusters which extend over wider regions. Due to these different length scales, it is then possible to obtain converged results on longer time scales. As already mentioned in Sec. 5, other types of expansions exist for the 2d square lattice (and have been used successfully for thermodynamic quantities) such as, e.g., clusters consisting of multiple corner-sharing $2 \times 2$ squares. We must admit, however, that it seems difficult a priori to estimate how good the convergence of the NLCE turns out for a specific choice of cluster types.

6) Indeed, for the (quasi)one-dimensional ladders, we only use rectangular clusters with a height that is equal to the number of the legs of the ladder. In the revised manuscript, we provide some clarifying discussion for this choice.

7) We thank the Referee for this comment. Concerning the notion of "topologically equivalent" clusters, we refer to our answer to point 4) above. Moreover, the Referee is absolutely correct that the Hamiltonian has to share the symmetries of the lattice in order to guarantee that clusters rotated by 90 degrees are equivalent. Overall, in the new manuscript, we have substantially revised the text in Section 3.1 (as well as the caption of Fig. 1) and hope that the details of the NLCE method and the related terminology are now clarified.

8) The Referee is of course correct that parity symmetry is present. We haven't mentioned this since (i) we actually did not use this symmetry in our simulations and (ii) the speed-up (or memory reduction) due to parity is considerably less compared to the more "powerful" U(1) and translational symmetries. In the revised manuscript, we now mention the parity symmetry.

9) We agree with the Referee that it would be helpful for the reader to compare to results for systems with OBC since the clusters within the NLCE are defined with OBC. While we were worried that the figures become too overloaded when adding further data for lower expansion orders and for systems with OBC, we have decided to add a new section in the Appendix where we show corresponding data in the new Fig. 9. As correctly expected by the Referee, Fig. 9 shows that finite-size corrections strongly depend on the choice of the boundary conditions. Namely, finite-size effects become apparent at earlier times for systems with OBC. In particular, comparing the convergence of these OBC data to corresponding NLCE results with C <= L, we find that the NLCE results remain converged for much longer times. In this sense, we conclude that NLCE is clearly advantageous compared to a direct simulation.

In addition to the new Fig. 9, we have also added data for lower expansion orders in Fig. 2(a) and Fig. 5.

10) There was actually no particular reason to omit the data for g = 1 in Fig. 3(b) other than trying not to overload the figure. In the revised manuscript, we have now added the data for g = 1. Concerning data for OBC, we have decided not to show them in Fig. 3 for the same reason. We hope that the Referee is satisfied with our solution to include the new Fig. 9 in the Appendix which shows exemplary data for systems with OBC and also NLCE results with lower expansion orders.

11) We agree with the Referee that the NLCE and the $5 \times 5$ simulation in Fig. 5 are essentially equally good. As we have already tried to argue in the old manuscript, we believe that a potential advantage of the NLCE certainly depends on the specific point in parameter space. Especially in cases where the quantity of interest exhibits only small finite-size effects, a straightforward calculation for a system of finite size might actually be a better choice. In contrast, if finite-size effects are pronounced, the NLCE can yield a real advantage. We have slightly extended our discussion around Fig. 5.

12) Since these are not our simulations and we are no experts on ANN, we are not in the position to make definite statements about the potentials of this method. At present, the digitized ANN data from Ref. [56] shown in Fig. 6 and Fig. 7 seem to be the state-of-the-art in terms of system sizes and time scales for this method.

13) We thank the Referee for this comment. Generally, if the self-consistent embedding of the clusters and subclusters is carried out correctly, then the restricted NLCE by construction should converge to the correct value (see e.g. Ref. [64] of the manuscript). For the particular case of the rectangle expansion on the 2D lattice used here, there can be some subtleties, e.g., when a new expansion order only adds a single chain (e.g., C = 29 would only add the cluster 29\times 1). In such a case, it might happen that these additional orders leave the series essentially unchanged, i.e., C = 28 and C = 29 agree very well (even at long times where the result is clearly wrong). In practice, it is therefore always helpful to look at multiple expansion orders, such as in Fig. 5 - 7 where we plot C = 24,27,28, and identify the position where the different orders deviate from each other.

We once again would like to thank the Referee for her/his comments which have helped us to improve our manuscript. Given our response and the changes made to the text, we hope that our manuscript is now ready for publication in SciPost Physics.

---

## Round 2 · Referee Report · Anonymous (Referee 3) · 2020-6-16

Strengths

State-of-the art numerical study of quantum quenches in spin models

Weaknesses

The authors should clarify certain aspects about the methodology

Report

The manuscript by Richter et al studies time evolution after quantum quenches in the transverse field Ising models in various lattice dimensions (chains, ladders and a square lattice). The main goal of the work is to compare two numerical techniques for unitary time evolution: the NLCE and exact diagonalization methods based on sparse matrices. The authors compare results case by case, and generally find that the NLCE is a competitive method that may provide in some parameter regimes very accurate predictions for time evolution of observables.

The manuscript is clearly written, the results are sound and of interest to a wide community of researchers studying nonequilibrium dynamics, quantum quenches, and developing computational methods. I therefore warmly recommend the paper for publication in SciPost Physics. It should be noted, though, that previous referees (reports 1 and 2) brought up relevant comments about the manuscript that should be answered before publication. The questions that I find particularly important are related to i) reasoning why rectangular clusters outperform other choices of clusters within the NLCE, ii) what are predictions of the NLCE if smaller orders of expansion are used, iii) to compare the NLCE results with exact diagonalization simulations with OBC.

Requested changes

Given by points i) - iii) in the report.

  • validity: high
  • significance: good
  • originality: good
  • clarity: high
  • formatting: excellent
  • grammar: good

Author:  Jonas Richter  on 2020-07-24  [id 907]

(in reply to Report 3 on 2020-06-16)

We thank this Referee very much for writing a report on our manuscript, as well
as for her/his overall positive assessment of our work.

In the revised manuscript, we have made various changes to the text and
to the figures which address the points i) - iii) requested by the Referee. For
more detailed replies to the individual points, we refer to our replies to the
reports by Referee 1 and Referee 2.

Given our response and the changes made to the text, we hope that our
manuscript is now ready for publication in SciPost Physics.

---

## Round 3 · Referee Report · Anonymous (Referee 1) · 2020-8-4

Report

The authors have made several interesting changes to the manuscript and have answered my previous questions and suggestions. I especially find the new Fig. 5 very interesting and the discussion about the different types of cluster expansions will be very instructive. It is also good to know, that a Lanczos time evolution could be used as well. The revised version has improved and so I can fully recommend publication in SciPost Physics now.

Requested changes

None

---

## Round 3 · Referee Report · Anonymous (Referee 2) · 2020-8-26

Report

The authors have clearly and satisfyingly addressed all my comments. With the changes and addition to their manuscript, it now reads very well and clearly demonstrates and discusses the advantages of the linked cluster expansion in quantum dynamics. The possible advantages and disadvantages of their method compared with other available methods is also clearly explained.

---

## Round 3 · Author Response

Dear Editors,

we thank you very much for the careful handling of our manuscript

"Quantum quench dynamics in the transverse-field Ising model: A numerical
expansion in linked rectangular clusters",

which we hereby would like to resubmit.

In the revised version, we have incorporated various changes to the
text and the figures according to the requested changes by the three Referees.
We believe that these changes further improve the quality and the readability
of our manuscript. Additionally, in our replies to the reports, we have
addressed all questions and comments by the Referees.

Given the overall positive reports of the three Referees, as well as our
changes made to the text, we hope that our manuscript is now ready for
publication in SciPost Physics.

Yours sincerely,

Jonas Richter,
Tjark Heitmann,
Robin Steinigeweg.

---

## Round 3 · List of Changes

Warnings issued while processing user-supplied markup:

  • Inconsistency: Markdown and reStructuredText syntaxes are mixed. Markdown will be used.
    Add "#coerce:reST" or "#coerce:plain" as the first line of your text to force reStructuredText or no markup.
    You may also contact the helpdesk if the formatting is incorrect and you are unable to edit your text.

Summary of Changes

  • various small changes throughout the text according to the comments by the Referees (as described in our replies to the Referee reports)

  • revision and extension of the explanations concerning the numerical method in Sec. 3.1

  • added new data for lower expansion orders in Fig. 2(b)

  • added new data for transverse field g = 1 in Fig. 3(b)

  • added new data for lower expansion order C = 10 in Fig. 5

  • updated digitized ANN data for a larger system size in Fig. 7

  • added a new section in the Appendix and the new Fig. 9, where additional data for lower expansion orders and systems with open boundary conditions are shown

  • updated all arXiv references with the published version where possible

---

## Editorial Decision

published